# Output Distribution over the Entire Input Space: A Novel Perspective to Understand Neural Networks

## Abstract

Understanding the input-output mapping relationship in the *entire input space* contributes a novel perspective to a comprehensive understanding of deep neural networks. In this paper, we focus on binary neural classifiers and propose to first uncover the histogram about the number of inputs that are mapped to certain output values and then scrutinize the representative inputs from a certain output range of interest, such as the positive-logit region that corresponds to one of the classes. A straightforward solution is uniform sampling (or exhaustive enumeration) in the entire input space but when the inputs are high dimensional, it can take almost forever to converge. We connect the output histogram to the *density of states* in physics by making an analogy between the energy of a system and the neural network output. Inspired by the Wang-Landau algorithm designed for sampling the density of states, we propose an efficient sampler that is driven to explore the under-explored output values through a gradient-based proposal. Compared with the random proposal in Wang-Landau algorithm, our gradient-based proposal converges faster as it can propose the inputs corresponding to the under-explored output values. Extensive experiments have verified the accuracy of the histogram generated by our sampler and also demonstrated interesting findings. For example, the models map many human unrecognizable images to very negative logit values. These properties of a neural model are revealed for the first time through our sampled statistics. We believe that our approach opens a new gate for neural model evaluation and shall be further explored in future works.

## 1 Introduction

Understanding the input-output mapping relationship in the *entire input space* contributes a novel perspective to a comprehensive understanding of deep neural networks. Existing methods approximate such mapping relations through the evaluation on a certain subset of the entire input space, such as measuring the accuracy on *in-distribution* test sets Dosovitskiy et al. (2021); Tolstikhin et al. (2021); Steiner et al. (2021); Chen et al. (2021); Zhuang et al. (2022); He et al. (2015), *out-of-distribution* (OOD) test sets (Liu et al., 2020; Hendrycks & Gimpel, 2016; Hendrycks et al., 2019; Hsu et al., 2020; Lee et al., 2017; 2018), and *adversarial* test sets Szegedy et al. (2013); Rozsa et al. (2016); Miyato et al. (2018); Kurakin et al. (2016). However, none of the existing evaluations can offer a comprehensive understanding that covers the entire input space, including all kinds of inputs mentioned above and even those *human unrecognizable* inputs as shown in Fig 1a.

As a pilot study, we focus on binary classification — given a trained binary classifier, we aim to uncover a histogram that counts how many samples in the entire input space are mapped to certain logit values, i.e., the distribution of the output values, as shown in Fig 1b. A straightforward solution is uniform sampling (or exhaustive enumeration) in the entire input space but when the inputs are high dimensional, it can take almost forever to converge. Therefore, it calls for a novel efficient sampling method over a neural model's output space. Note that, as a side product of the sampling procedure, one can expect that this histogram also offers fine-grained information such as some representative input samples corresponding to a certain range of output values.

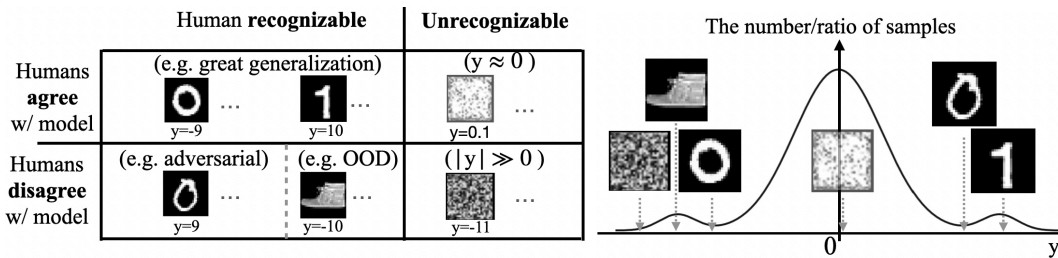

(a) Different types of input samples    (b) The histogram of an example binary classifier

Figure 1: Input types and the example output histogram when the task is binary classification between digits 0 and 1. The entire input space covers all possible gray-scale images of the same shape. y is the output (logit) of input x.

We connect the output histogram problem to the *density of states* (DOS) problem in physics by making an analogy between the system energy and neural network output, as shown in the right figure. If one follows the physics language to describe our problem, the input $\mathbf{x}$ to the neural network can be viewed as the configuration $\mathbf{x}$ of the system; the neural network output (e.g., logit values in binary classifier) $y(\mathbf{x})$ corresponds to the energy function $E(\mathbf{x})$; the desired output histogram can be obtained through the DOS (a.k.a., the entropy, $S(E(\mathbf{x}))$, the log scale of DOS), which is the count of the configurations given the energy value. Note that the density of states by definition is over the entire input space, which aligns perfectly with our objective.

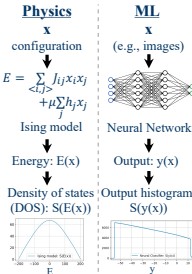

Inspired by the Wang-Landau algorithm (Wang & Landau, 2001) designed for density of states, we propose an efficient sampler that is driven to explore the under-explored output values through a gradient-based proposal. If the binary classifier is well-trained, it is reasonable to believe that in-distribution inputs (images) which usually have certain semantic structures are concentrated in certain output ranges. If one follows the random proposal in the Wang-Landau algorithm, it is difficult to propose the inputs with meaningful structure (or even in-distribution inputs), thus possibly preventing the sampler from exploring the corresponding output values. Thus, we propose to apply a gradient-based proposal called Gibbs-with-Gradients (GWG) (Grathwohl et al., 2021) which proves to be efficient to propose in-distribution inputs for a trained model.

With the help of this new sampler, we can reveal some new understanding of the models for the entire input space. First, in our experiments on a real-world dataset, the dominant output values are very negative and correspond to the human-unrecognizable inputs. This indicates the models may map an overwhelmingly large number of unrecognizable images to the overconfident prediction probabilities. Second, we can derive the relative difference between the dominant peak of output values and the other output values, especially those where the in-distribution inputs correspond to. The output values where the in-distribution inputs correspond to are also dominated by the human-unrecognizable inputs. This result presents significant challenges to the OOD detection problems. Third, we observe a clear trend of the representative samples in a CNN model and speculate it simply utilizes the background to predict the labels of the digits.

Our contributions are summarized as follows.

- We work on the challenging problem to uncover the output distribution over the entire input space. Such output distributions offer a novel perspective to understand deep neural networks.
- We connect this output distribution problem to the density of states problem in physic and successfully tailor Wang-Landau algorithm using a gradient-based proposal, which is a must-have component to sample the entire output space as much as possible, improving the efficiency.
- We conduct extensive experiments on toy and real-world datasets based on CNN and ResNet-18 to confirm the correctness of our proposed sampler and discover novel and interesting findings.

We believe that our approach opens a new gate for neural model evaluation and shall be further explored in future works. For example, one can can utilize our sampler to estimate the intrinsic *ratio* of in-distribution samples given a range of interest with human evaluation as shown in Sec. 5.3.

## 2 PROBLEM DEFINITION

In the traditional setting, binary neural classifiers model the class distribution through logit $z$. A neural classifier parameterized by $\theta$ learns $p_\theta(z|\mathbf{x}) = \delta(z - y_\theta(\mathbf{x}))$ through a function $y_\theta : \mathbf{x} \to z \in \mathbb{R}$, where $\mathbf{x} \in \Omega$, $\Omega \subseteq \{0, ..., N\}^D$ for images, and $\delta$ is the Dirac delta function. $\Omega$ is aligned with Gibbs-With-Gradient's setting to be discrete.

What the above model does not define is the distribution of the data $\mathbf{x}$. This paper aims to obtain the output value distribution of binary classifiers in the entire input space: $\Omega = \{0, ..., N\}^D$. Here we assume that the data distribution $p(\mathbf{x})$ follows the uniform distribution over the domain $\Omega$ of $\mathbf{x}$ and denote its measure by $\mu$. We define the joint distribution

$$p_\theta(z, \mathbf{x}) = p_\theta(z|\mathbf{x})\mu(\mathbf{x})$$

Our goal is only the logit (output) distribution. We marginalize the above joint distribution to define the density given the logit $z$:

$$p_\theta(z) = \sum_\Omega p_\theta(z|\mathbf{x})\mu(\mathbf{x}) = \sum_{\mathbf{x} \in \Omega} \delta(z - y_\theta(\mathbf{x}))$$

To sample from the distribution $p_\theta(z)$, we can first sample $\mathbf{x}_i \sim \text{Uniform}(\Omega)$, then condition on the sampled $\mathbf{x}_i$, sample $z_i \sim p_\theta(z|\mathbf{x}_i)$. While uniform sampler in principle can resolve our problem, it takes almost forever to converge.

## 3 METHOD

In this section, we first discuss the connection between our problem to density of states (DOS), introduce both Wang-Landau algorithm and Gibbs-with-Gradient as background, and present our new sampler Gradient-Wang-Landau algorithm.

### 3.1 CONNECTION TO DENSITY OF STATES (DOS) IN PHYSICS

In statistical physics, given the energy function $E : \mathbf{x} \to \mathcal{E} \in \mathbb{R}$ , the DOS $\rho(\mathcal{E})$ is defined as

$$\rho(\mathcal{E}) = \sum_{\mathbf{x} \in \Omega} \delta(\mathcal{E} - E(\mathbf{x}))$$

where $\delta$ is the Dirac delta function and $\Omega$ is the domain of $\mathbf{x}$ where $\mathbf{x}$ is valid. The DOS is treated as a probability distribution in the energy space, whose log-probability is defined as the entropy $S$:

$$\rho(\mathcal{E}) = \exp(S(\mathcal{E}))$$

Boltzmann constant is assumed to be 1 in our setting. DOS is meaningful because many physical quantities depend on energy or its integration but not the specific input $\mathbf{x}$.

We connect the output histogram to DOS in physics by making an analogy between the system energy $\mathcal{E} = E(\mathbf{x})$ and neural network output $z = y(\mathbf{x})$. This connection is based on the observation that the energy function in physics maps an input configuration to a scalar-valued energy; similarly, a binary neural classifier maps an image to a logit. Both the logit and energy are treated as the direct output of the mapping. Other quantities, such as the loss, are derived from the output. The desired output histogram can be obtained similarly through sampling the DOS (a.k.a., the entropy $S(E(\mathbf{x}))$ or $S(y(\mathbf{x}))$ in the log scale) which is the count of the configurations given the energy value. The output histogram and DOS are defined in the entire input space.

### 3.2 WANG-LANDAU ALGORITHM AND GIBBS-WITH-GRADIENT

**Wang-Landau algorithm** is a Markov chain Monte Carlo sampler that samples DOS. Since the true distribution $\rho(\mathcal{E})$ is what we are interested in sampling but its formula/model is unknown, we need to approximate it. The Wang-Landau algorithm uses a histogram to store the current estimation $\tilde{S}$. It improves the sampling efficiency by sampling the inverted distribution:

$$p(\mathbf{x}) \propto \exp(-\tilde{S}(E(\mathbf{x})))$$

By sampling $p(\mathbf{x})$, we can get an ensemble of $\mathcal{E}$ via $E(\cdot)$ whose probability distribution is:

$$\pi(\mathcal{E}) = \sum_{\mathbf{x} \sim p(\mathbf{x})} \delta(\mathcal{E} - E(\mathbf{x}))$$

When $\tilde{S}(\mathcal{E})$ approaches $S(\mathcal{E})$, the energy distribution $\pi(\mathcal{E})$ approaches to $\mathcal{E}$-independent constant for all the accessible energy $\mathcal{E}$.

**Gibbs-With-Gradient (GWG)** is used for energy-based models (EBM) by sampling

$$\log p(\mathbf{x}) = f(\mathbf{x}) - \log Z,$$

where $f(\mathbf{x})$ is the unnormalized log-probability, $Z$ is the partition function, and $\mathbf{x}$ is discrete. Typical Gibbs sampler iterates every dimension $x_i$ of $\mathbf{x}$, computes the conditional probability $p(x_i | x_1, ... x_{i-1}, x_{i+1}, ..., x_D)$, and samples according to this conditional probability.

When the training data $\mathbf{x}$ are natural images and the EBM learns $\mathbf{x}$ decently well, the traditional Gibbs sampler wastes much of the computation. For example, most pixel-by-pixel iterations over $x_i$ in MNIST dataset will be on the background which should stay black. GWG proposes a smart proposal that picks the pixel $x_i$ that is more likely to change, such as the pixels around the edge between the bright and dark region of the digits.

### 3.3 WANG-LANDAU WITH GRADIENT PROPOSAL

Directly applying Wang-Landau algorithm is not enough as it uses random proposal, because a trained neural model learns preferred mapping through the loss function. For example, a binary classifier should map the training inputs to either the sufficiently positive or negative logit values which ideally should correspond to the extremely rare but semantically meaningful inputs. After the sampler explores and generates the peak centered at 0 where most random samples correspond to as shown in Fig. 1b, it is almost impossible for the sampler with a random proposal to propose the inputs with meaningful structure (or even in-distribution inputs) so that the other possible output values are explored. Of course, whether those output values correspond to in-distribution inputs is only confirmable after sampling. In summary, it is extremely difficult for the random proposal in Wang-Landau algorithm to explore (almost all) the possible output values.

We propose to use the framework of Wang-Landau algorithm but replace the proposal distribution with the Gibbs-With-Gradients (GWG) sampler which has a gradient proposal, since the gradient proposal takes the advantage of model's learned weights to propose inputs. In order to sample the distribution of the output prediction through GWG, we define log-probability $f(x)$ as:

$$f(x) = S(y(\mathbf{x}))$$

where $S$ is the count for the bin corresponding to $y(\mathbf{x})$. The fixed $f(\cdot)$ in the original GWG is changing in our sampling process given the input $\mathbf{x}$, since the formula for $S$ is unknown and we can only estimate the output distribution as we did in Wang-Landau algorithm. Moreover, the GWG requires the gradient of $f$, but the $S$ is not differentiable since it is approximated through discrete bins. We adopt a first-order differentiable interpolation for the discrete histogram of entropy.

In summary, similar to the original Wang-Landau algorithm, we first initialize two histograms with all of their bins to 0. One of these histograms is for entropy $S$, and the other histogram, $H$, is a counter of how many times the sampler workers visited a specific bin and $H$ is also for the flatness check. We first preset the number of iterations. When $H$ passes the flatness check, it enters the next iteration loop with the step counter reset to 0. Every step in the while loop until the flatness check passes, we interpolate the entropy in the histogram to get a differentiable interpolation and take the derivative of the negation of the entropy with respect to the output $z$ and the inputs $\mathbf{x}$ through the chain rule. GWG uses this gradient to propose the next input that is likely to have *lower* entropy and be accepted by the sampler. his procedure drives the sampler to visit rare samples whose logit values correspond to the lower entropy until $S$ converges. This proposal also goes through a Monte-Carlo accept-reject procedure in the GWG. Once the flatness is met, the bins of $H$ are reset to 0 and the step size is halved before a new iteration. Our proposed algorithm is in Alg. 1 in Appendix.

## 4 RELATED WORKS AND DISCUSSIONS

**Performance Characterization** has long been explored even before the era of deep learning (Haralick, 1992; Klette et al., 2000; Thacker et al., 2008). The input-output relationship has been explored for simple functions (Hammitt & Bartlett, 1995) and mathematical morphological operators (Gao et al., 2002; Kanungo & Haralick, 1990). Compared to existing performance characterization approaches (Ramesh et al., 1997; Bowyer & Phillips, 1998; Aghdasi, 1994; Ramesh & Haralick, 1992; 1994), our work focuses on the output distribution (Greiffenhagen et al., 2001) of a neural network over the entire input space (i.e., not task specific) following the blackbox approach (Courtney et al., 1997; Cho et al., 1997) where the system transfer function from input to output is unknown. Our setting shall be viewed as the most general forward uncertainty quantification case (Lee & Chen, 2009) where the model performance is characterized when the inputs are perturbed (Roberts et al., 2021). To our best knowledge, we demonstrate for the first time that the challenging task of sampling the entire input space for modern neural networks is feasible and efficient by drawing the connection between neural network and physics models. Our proposed method can offer samples to be further integrated with the performance characterization methods mentioned above.

**Density Estimation and Energy Landscape Mapping** Previous works in density estimation focus on data density Tabak & Turner (2013); Liu et al. (2021), where class samples are given and the goal is to estimate the density of samples. Here we are not interested in the density of the given dataset, but the density of all the valid samples in the pixel space for a trained model. Hill et al. (2019); Barbu & Zhu (2020) have done the pioneering work in sampling the energy landscape for energy-based models. Their methods specifically focus on the local minimum and barriers of the energy landscape. We can relax the requirement and generalize the mapping on the "output" space where either sufficiently positive or sufficiently negative output (logit) values are meaningful in binary classifiers and other models.

**Open-world Model Evaluation** Though many neural models have achieved the SOTA performance, most of them are only on in-distribution test sets (Dosovitskiy et al., 2021; Tolstikhin et al., 2021; Steiner et al., 2021; Chen et al., 2021; Zhuang et al., 2022; He et al., 2015; Simonyan & Zisserman, 2014; Szegedy et al., 2015; Huang et al., 2017; Zagoruyko & Komodakis, 2016). Open-world settings where the test set distribution differs from the in-distribution training set create special challenges for the model. While the models have to detect the OOD samples from in-distribution samples (Liu et al., 2020; Hendrycks & Gimpel, 2016; Hendrycks et al., 2019; Hsu et al., 2020; Lee et al., 2017; 2018; Liang et al., 2018; Mohseni et al., 2020; Ren et al., 2019), we also expect sometimes the model could generalize what it learns to OOD datasets (Cao et al., 2022; Sun & Li, 2022). It has been discovered that models have over-confident predictions for some OOD samples that obviously do not align with human judgments (Nguyen et al., 2015). The OOD generalization becomes more challenging because of this discovery, because the models may not be as reliable as we thought they were. Adversarial test sets Szegedy et al. (2013); Rozsa et al. (2016); Miyato et al. (2018); Kurakin et al. (2016); Xie et al. (2019); Madry et al. (2017) also present special challenges as models decisions are different from those of humans. Having a full view of input-output relation with all the above different kinds of test sets under consideration is important.

**Samplers** MCMC samplers (Chen et al., 2014; Welling & Teh, 2011; Li et al., 2016; Xu et al., 2018) are developed to scale to big datasets and sample efficient with gradients. Recently, Gibbs-With-Gradients (GWG) (Grathwohl et al., 2021) is proposed to pick the promising pixel(s) as the proposal. To further improve sampling efficiency, CSGLD (Deng et al., 2020) drives the sampler to explore the under-explored energy using similar idea as Wang-Landau algorithm (Wang & Landau, 2001). The important difference between our problem setting and the previous ones solved by other MCMC samplers is the function or model as distribution to be sampled from is unknown. Wang-Landau algorithm utilizes previous approximation of the distribution to drive the sampler to explore the under-explored energy regions. This algorithm can be more efficient through parallelization (Vogel et al., 2013; Cunha-Netto et al., 2008), bin-free (Junghans et al., 2014; Li & Eisenbach, 2017) and extended to multi-dimensional outputs (Zhou et al., 2006). While the previous samplers can be applied to high dimensional inputs, the energy functions written by physicists are relative simple and symmetric. However, modern neural networks are complex and hard to characterize performance (Roberts et al., 2021). We assume agnostic of the output properties of the model and thus apply the Wang-Landau algorithm to sample the entropy as a function of energy but with the gradient proposal in GWG to make the sampler more efficient. Similar to GWG, our sampler can propose

the inputs corresponding to the under-explored regions of outputs. Improvements of efficiency can benefit from a patch of pixel changes.

## 5  EXPERIMENTS

In this section, we apply our proposed Gradient Wang-Landau sampler to inspect a few neural network models and present the discovered output histogram together with representative samples. The dataset and model training details are introduced in Sec. 5.1. We first empirically confirm our sampler performance through a toy example in Sec. 5.2. We then discuss results for modern binary classifiers in Sec. 5.3 and Sec. 5.4. Hyperparameters of the samplers tested in are Appendix C.

### 5.1  DATASETS, MODELS, AND OTHER EXPERIMENT SETTINGS

**Datasets**    As aforementioned, we focus on binary classification. Therefore, we derive two datasets from the MNIST datasets by only including samples with labels $\{0, 1\}$. The training and test splits are the same as those in the original MNIST dataset.

- **Toy** is a simple dataset with $5 \times 5$ binary input images we construct. It is designed to make feasible the bruteforce enumeration over the entire input space (only $2^{5 \times 5}$ different samples). We center crop the MNIST samples from $\{0, 1\}$ classes and resize them to $5 \times 5$ images. We compute the average of the pixel values and use the average as the threshold to binarize the images — the pixel value lower than this threshold becomes $0$; otherwise, it becomes $1$. The duplicates are not removed for accuracy after resizing since PyTorch does not find duplicate row indices.
- **MNIST-0/1** is an MNIST dataset whose samples only have the 0,1 labels. To align with the GWG setting, the inputs are discrete and not Z-normalized. Therefore, in this dataset, the input $\mathbf{x}$ is $28 \times 28$ dimensional with discrete pixel values from $\{0, ..., 255\}$.

**Neural Network Models for Evaluation**    Since the focus of this paper is not to compare different neural architectures, given the relatively small datasets we have, we train two types of models, a simple CNN and **ResNet-18** (He et al., 2015). Each pixel of the inputs is first transformed to the one-hot encoding and passed to a 3-by-3 convolution layer with 3 channel output. The **CNN** model contains 2 convolution layers with 3-by-3 filter size. The output channels are 32 and 128. The final features are average-pooled and passed to a fully-connected layer for the binary classification.

Please keep in mind that our goal in this experiment section is to showcase that our proposed sampler can uncover some novel interesting empirical insights for neural network models. Models with different architectures, weights due to different initialization, optimization, and/or datasets will lead to different results. Therefore, our results and discussions are all *model-specific*. Specifically, we train a simple CNN model to classify the $5 \times 5$ binary images in the Toy dataset (**CNN-Toy**). The test accuracy of this CNN-Toy model reaches $99.7\%$, which is almost perfect. We train a simple CNN model to classify the $28 \times 28$ grey-scale images in the MNIST-0/1 dataset (**CNN-MNIST-0/1**). The test accuracy of CNN-MNIST-0/1 model is $97.8\%$. We train a ResNet-18 model to classify the $28 \times 28$ grey-scale images in the MNIST-0/1 dataset (**ResNet-18-MNIST-0/1**). The test accuracy of ResNet-18-MNIST-0/1 model is $100\%$.

**Sampling Methods for Comparison**    We compare several different sampling methods (including our proposed method) to obtain the output histogram over the entire input space.

- **Enumeration** generates the histogram by enumerating all the possible pixel values as inputs. This is a rather slow but the most accurate method.
- **In-dist Test Samples** generates the histogram of the inputs based on the fixed test set. This is commonly used in machine learning evaluation. It is based on a very small and potentially biased subset of the entire input space.
- Wang-Landau algorithm (**WL**) generates the histogram the Wang-Landau algorithm with the random proposal. Specifically, we randomly pick one pixel at a time and change it to any valid (discrete) value as in this implementation [1].
- Gradient Wang-Landau (**GWL**) generates the histogram by our proposed sampler of Wang-Landau algorithm with gradient proposal.

---

[1] `https://www.physics.rutgers.edu/~haule/681/src_MC/python_codes/wangLand.py`

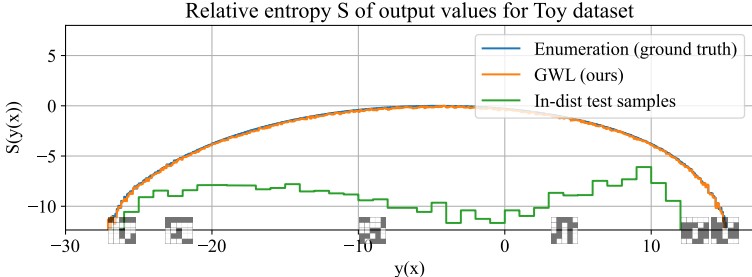

Figure 2: Output histograms of CNN-Toy obtained by different sampling methods. The in-distribution samples are only a very small portion in the output histogram. We also present the representative samples obtained by GWL given different logit values.

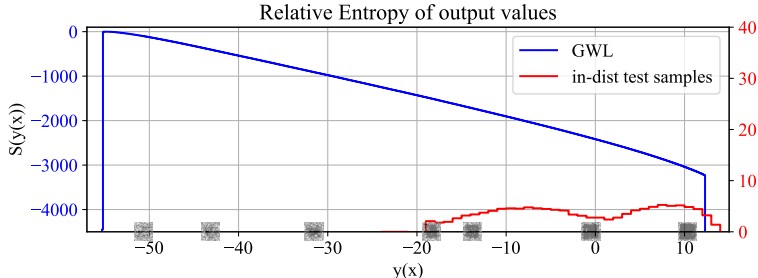

Figure 3: Output histograms of CNN-MNIST-0/1 obtained by different sampling methods. The blue scale is for GWL and the red scale is for In-distribution Test Samples. We also present the representative samples obtained by GWL given different logit values (more in Fig. 6 in Appendix)

## 5.2 RESULTS OF CNN-TOY

Given the CNN-Toy model, we apply Enumeration, GWL, and In-dist Test Samples to obtain the output histograms, as shown in Fig. 2. Note that our GWL method samples the relative entropy of different energy values as duplicate **x** may be proposed. After normalization with the maximum entropy, the GWL histogram almost exactly matches the Enumeration histogram which is the ground truth histogram. This confirms the accuracy of our GWL sampler and we can apply it further to more complicated models with confidence.

Remarkably, this histogram is quite different from the expectation we presented in Fig. 1b — this histogram is even not centered at 0 or has the expected subdominant peaks on both the positive and negative sides. Instead, the dominant peak is so wide that it covers almost the entire spectrum of the possible output values. From a coarse-grained overview, most of the samples are mapped to the center of logit $-5$ with a decay from $-5$ to both sides in the CNN-Toy model. This shows the CNN-Toy model is biased to predict more samples to the negative logit values.

In Fig. 2, we also present the representative samples obtained by GWL given different logit values in the CNN-Toy model. The visualization results suggest that the CNN-Toy model probably learns the digit "1" for positive logit values as the center pixels of the representative samples are white (see the three representative samples with logit values from 0 to 20) and "0" for the very negative logit values as the center pixels of the representative samples are black (see two representative samples with logit values from -20 to -30). From this example, one can see that the output histogram over the entire input space can offer a comprehensive understanding of the neural network models, helping researchers better understand critical questions such as the distribution of the outputs, where the model maps the samples to, and what the representative samples with high likelihood are.

## 5.3 RESULTS OF CNN-MNIST-0/1

**Histogram Results by GWL** The trial of applying GWL on the CNN-Toy model is encouraging and we now apply GWL to the CNN-MNIST-0/1 that is trained on a real-world dataset. The results are shown in Fig. 3. As our GWL reveals, the output histogram of CNN-MNIST-0/1, similar to CNN-Toy's histogram, does not have the subdominant peaks. It is also different from the presumed case in Fig. 1b. Compared with the output histogram of the CNN-Toy model (i.e., Fig. 2), this

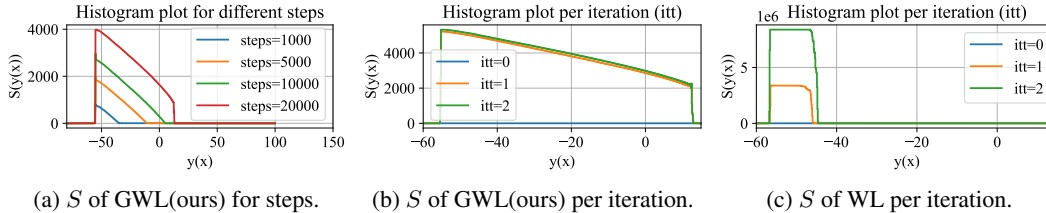

(a) $S$ of GWL(ours) for steps.  (b) $S$ of GWL(ours) per iteration.  (c) $S$ of WL per iteration.

Figure 4: Intermediate output histogram $S$ per iteration. (a) GWL gradually explores the logit values in the first iteration. (b) GWL discovers the output histogram well within 2 iterations. (c) The original WL explores the output distribution much slower.

time, the peak is on the negative boundary and the histogram is skewed towards the negative logit values. $S$ almost linearly decays to the positive logit values. While the in-distribution samples have logit values between $-20$ and $15$ as we expect, these samples are exponentially (i.e., $e^{1300}$ at logit value -20 to $e^{3100}$ at logit value 13, thousands in log scale) less often found than the majority samples whose logit values are around $-55$. From a fine-grained view, the CNN-MNIST-0/1 model tends to map the human-unrecognizable samples to the very negative logit values. While previous work (Nguyen et al., 2015) showed the existence of the overconfident prediction samples, our result shows a rough but quantitative performance of this CNN which can serve as a baseline for further improvements. One may notice that in Fig. 3, there is still some output values (e.g., the rightmost positive logit region) that are not yet covered by our GWL sampler. We believe that this calls for more future work to follow on more advanced efficient samplers.

**GWL is much more efficient than WL**  Since WL takes a much longer time to converge, we are not able to obtain the converged results from WL. For the comparison purpose, we inspect the intermediate $S$ results of the GWL and WL samplers, as shown in Fig. 4. As one can see from Fig. 4a, in the first iteration, GWL has already been able to gradually explore the logit values efficiently from the most dominant output value around $-55$ to the positive logit values. Within only two iterations, as shown in Fig. 4b, GWL can discover the output histogram covering the value range from $-55$ to $13$. On the other hand, as presented in Fig. 4c, in the first two iterations, the original WL can only explore the output ranges from around $-55$ to $-45$; in the 3rd iteration, WL converges significantly slower and never ends in a reasonable time. This result indicates that the GWL converges much faster than the original WL and is able to explore a much more diverse range of output values.

**Manual inspection on more representative samples**  As show in Fig. 3, for the CNN-MNIST-0/1 model, GWL can effectively sample input images from logit values ranging from -55 to 13. We further group these logit values per 100 bins (100 bins correspond to a difference of 10 in logit value) in $S$, resulting in about 7 groups. For every group, we sample 200 representative input images. To make sure they are not correlated, we sample every 1000 steps. For demonstration purposes, we randomly pick 10-out-of-200 samples from every group in Fig. 6 in Appendix. We manually inspect the sufficiently positive group (e.g., the last column in Fig. 6) and the sufficiently negative groups (e.g., the first five columns in Fig. 6) , and there are no human recognizable samples of digits. We also observe an interesting pattern that as the logit value increases, more and more representative samples have black background. This result suggests that the CNN-MNIST-0/1 model may heavily rely on the background to classify the images (Xiao et al., 2020). We conjecture that is because the samples in the most dominant peak are closer to class 0 samples than class 1 samples (Appendix. D). In summary, although CNN-MNIST-0/1 holds a very high in-distribution test accuracy, it is far from a robust model because it does not truly understand the semantic structure of the digits.

**Discussion**  Fig. 3 presents challenges to the OOD detection methods that may be more model-dependent than we thought before. If the model cannot map most of the human unrecognizable samples with high uncertainty, the likelihood-based OOD detection methods (Liu et al., 2020; Hendrycks & Gimpel, 2016) cannot perform well for samples in the entire input space. Fig. 6 shows the inputs with the in-distribution output values (output logits of the red plot) of the CNN model may not uniquely correspond to in-distribution samples. More rigorous experiments to a definite conclusion are yet required as future work.

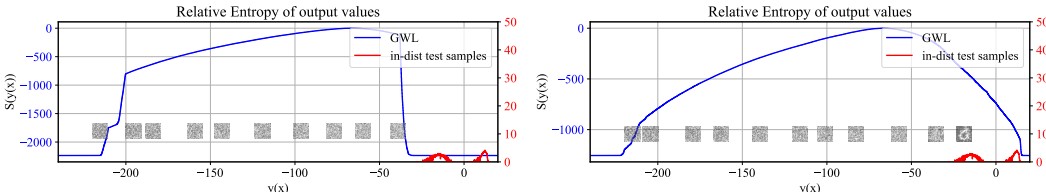

(a) Results with random re-initialization.  (b) Results with test set re-initialization.

Figure 5: Output histograms of ResNet-18-MNIST-0/1 obtained by different sampling methods. There may be a sharp local minima in the output landscape causing a cliff around the logit value of -33 and making GWL "trapped". We have tried two variants to address the "trapped" issue via (a) random re-initialization and (b) test set re-initialization. The blue scale is for GWL and the red scale is for In-distribution Test Samples. We also present the representative samples obtained by GWL given different logit values.

## 5.4 RESULTS OF RESNET-18-MNIST-0/1

When applying our GWL samplers to the ResNet-18-MNIST-0/1 model, we observe that the sampler can easily get trapped in some output regions. This is a fairly common phenomenon for Wang-Landau-based samplers, as reported in (Vogel et al., 2018). We follow the common practice to re-initialize the sampler to the random samples every time it gets trapped. We let those workers run 1000 steps (10,000 pixels selected with replacement) before counting to $S$ again. As shown in Fig. 5a, the smallest logit values in ResNet-18-MNIST-0/1 are around -220, much lower than those of CNN-MNIST-0/1. A wide range of negative logit values corresponds to human unrecognizable inputs and there is no obvious pattern observed in contrast to CNN-MNIST-0/1's results.

Interestingly, we observe a cliff around the logit value of -33. We try another variant for re-initialization: we re-initialize using the test set samples every time it gets trapped in certain output value. This time, as shown in Figure 5b, it can explore the output values larger than -33. We believe there may be a sharp local minima in the output landscape, similar to the case discussed before Vogel et al. (2018).

Because of the "trapping" issue and the complexity of ResNet-18 over CNN, we have to relax the flatness check a bit to let GWL converge for the first iteration. Because of the less rigorous flatness check, we do not draw conclusions about ResNet-18-MNIST-0/1 evaluation of the relative entropy differences. Compared with the CNN-MNIST-0/1 model, ResNet-18-MNIST-0/1 has more interesting phenomena and further exploration is needed to understand these phenomena.

## 6 CONCLUSION

We aim to get a full picture of the input-output relationship of a model through the inputs valid in the pixel space. We propose to use a histogram to better understand the input-output distribution. When the inputs are high-dimensional, enumeration or uniform sampling is either impossible or takes too long to converge. We connect the density of states in physics to this histogram sampling problem. We propose to use an efficient sampler to achieve this goal. We confirm empirically this can be achieved and uncover some new aspects of neural networks.

For future work, it is interesting to develop a new and more efficient sampler that has theoretical guarantees to acquire this input-output relationship in order to sample with more pixels, such as the ImageNet (Deng et al., 2009). Most importantly, with this new sampler, we can develop new insights into network architectures developed in the last decade for *open-world* applications.

## 7 REPRODUCIBILITY

We provide fairly amount of information to re-implement our sampler. The data processing is in the Sec. 5.1 and algorithm is in Appendix B. The hyperparameters and sampling details are also listed in the Sec. 5. We also provide different time stamp of steps for our samplers to indicate what to expect during the sampling procedure in Fig. 4. Of course, the Wang-Landau algorithm we adopted

is the prototypical one and it subjects to some issues reported in its follow-up works, such as the discontinuity of the boundaries between bins and trapping in one of the bins. These problems lead to some issues in our experiments and we discussed them in Sec. 5.4. More advanced algorithms have been developed to resolve these issues.

## 8 ETHICS STATEMENT

Our method aims to provide a comprehensitve understanding of the neural models. This work will be applicable to many applications, such as those in the safety and trusty-worthy machine learning. As a pilor study, we do not anticipate the negative aspects of our work.

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

## A APPENDIX A: REPRESENTATIVE INPUTS

Here we list more representative samples of the CNN-MNIST-0/1 scenario. The samples are bounded by a black box of boundaries.

| -50.6 | -43.2 | -31.5 | -18.3 | -13.8 | -0.5 | 10.3 |
| -54.2 | -43.9 | -28.9 | -18.5 | -13.9 | 0.6 | 8.8 |
| -46.7 | -44.2 | -32.8 | -17.4 | -13.9 | -3.5 | 11.2 |
| -49.3 | -39.0 | -26.0 | -20.2 | -9.8 | 0.7 | 7.9 |
| -49.1 | -39.6 | -33.4 | -23.4 | -11.0 | 0.8 | 6.3 |
| -47.9 | -35.7 | -32.2 | -18.6 | -7.6 | -0.7 | 9.3 |
| -47.2 | -43.1 | -33.7 | -16.2 | -14.9 | 3.4 | 10.4 |
| -45.1 | -38.1 | -28.8 | -23.4 | -12.2 | 3.9 | 14.8 |
| -53.1 | -36.7 | -33.5 | -22.5 | -5.7 | -1.9 | 8.8 |
| -48.7 | -43.8 | -31.7 | -22.2 | -8.0 | 0.2 | 5.8 |

Figure 6: More representative samples of the CNN-MNIST-0/1 model obtained by GWL at different logit values, grouped by logit values. We further group these logit values per 100 bins (100 bins correspond to a difference of 10 in logit value) in $S$, resulting in about 10 groups. The output values in the first column are within the range [-55,-45) and the second is from [-45,-35], etc.

## B APPENDIX B: GRADIENT WANG-LANDAU ALGORITHM

Here we provide the algorithms of the GWL algorithm. The input and output are listed. The hyper-parameters are determined mostly by the toy-example.

---

**Algorithm 1** Our proposed Gradient Wang-Landau (**GWL**)

---

**Require:** pretrained model $y$: $\mathbf{x} \rightarrow z$, flat histogram $H = 0$, entropy histogram $S = 0$, increment/step-size lnf, number of iterations T, test set $\mathcal{D}_{te}$, GWG sampler $GWG(z, S)$, interpolation function $g(z, S)$

  **for** $t = 1, 2, ..., T$ **do**
    $\mathbf{x} \sim \mathcal{D}_{te}$
    **while** H is not flat **do**
      $z = y(\mathbf{x})$
      $S_{in} = g(z, S)$              ▷ Get the continuous interpolation entropy $S_{in}$ at output $z$
      $\mathbf{x} \sim GWG(z, -S_{in})$      ▷ Take the negation of $S_{in}$ for Wang-Landau exploration
      $\tilde{z} = \text{round}(z)$      ▷ Round $z$ to the nearest $z'$ that corresponds to one of the bins
      $S[\tilde{z}] \leftarrow S[\tilde{z}]+\text{lnf}$
      $H[\tilde{z}] \leftarrow H[\tilde{z}]+1$
    **end while**
    lnf $\leftarrow$ lnf/2
    $H \leftarrow 0$                 ▷ Reset all the bins in counter $H$ to 0
  **end for**
  **return** $S(y)$

---

## C HYPER-PARAMETERS AND IMPLEMENTATION DETAILS FOR GWL AND WL

The hyper-parameters for GWL and WL are extremely similar, if not identical, as the only major difference between GWL and WL is the gradient proposal versus the random proposal. We first preset a large enough range of output values for the sampler to explore the trained neural network models. In our experiments, we found that the output (logit) values of the binary classifiers typically fall in the range of -300 to 100 (based on ResNet). Therefore, we use this range for all experiments. For flatness histogram $H$, the bin window size is set to be 1, resulting in 400 bins. The histogram $H$ is considered flat if the difference between maximum bin value and minimum bin value is smaller than the average bin value. For output histogram $S$, we set the bin window size to be 0.1, resulting in 4000 bins. Instead of updating one bin at a time for $S$, we update the neighbor bins with exponential decay. We use the linear interpolation to approximate the bins for continuous queries. We iterate 5 times with test set initialization. Every step the GWG tries to at most update 10 pixels.

## D SAMPLES SIMILARITY

The samples in the most dominant peak may be closer to class 0 than to class 1. We compute the L2 pixel-wise distance from the uniform noise image to the samples of class 1 and 0 respectively. The mean L2 distance from uniform noise to 0 is around 0.3121 and that from uniform noise to 1 is around 0.3236. The distance between 1 and 0 samples is 0.1652. This result shows the samples in the most dominant peak are closer to class 0 samples than class 1 samples. More rigorous experiments to a definite conclusion is yet required as future work.

