# OpenReview forum: "Output Distribution over the Entire Input Space: A Novel Perspective to Understand Neural Networks"
_ICLR.cc/2023/Conference — Submitted to ICLR 2023_

### Official Review · Reviewer_ucC6 · 2022-10-24

**Confidence:** 2
**Correctness:** 3
**Technical Novelty And Significance:** 2
**Empirical Novelty And Significance:** 2
**Recommendation:** 6

**Clarity, Quality, Novelty And Reproducibility:**

The paper is clearly written. There is novelty in the proposed algorithm. No artefacts were made available for reproducibility.


**Strength And Weaknesses:**

-= S1 =- The analogy that the paper draws between neural networks' output histogram and the density of states problem in physics is well-motivated. This open opportunities to apply established methods from physics to the domain of  neural networks.

-= S2 =- The authors make the observation that the randomized sampling method of WL is not as applicable in the context of neural networks and modify that by adding a gradient-based approach that includes more structured inputs.

-= W1 =- The paper considers a very limited set of neural network models that are not state-of-the-art. The paper can benefit from using a recent architecture to showcase the utility of the approach. The same applies to datasets. It is unclear why binary classification on MNIST is an interesting case to apply this technique to.

-= W2 =- It is not sufficiently clear how the new insights uncovered by applying this technique augment our understanding of neural networks. The paper can benefit from spelling that out more explicitly.

-= W3 =- It is also unclear how this approach can be extended to the case of multiple labels. The paper can benefit from a discussion on that aspect as >2 label classification is more prevalent than a binary classifier.


**Summary Of The Paper:**

The paper addresses the problem of efficiently understanding the relationship between the input-output mapping of a neural network over a large and representative set of input spaces. To do so, the authors draw an analogy between a neural network model and physics's density of states problem. In doing so, they can design a new sampling strategy inspired by the Wang-Landau algorithm. They augment the algorithm with a gradient-based approach that can sample inputs with maeaningful structure. They conduct experiments on ResNet and CNN to show the approach's effectiveness.


**Summary Of The Review:**

While the paper is well-motivated and the algorithm proposed is relatively novel, the experiments are limited in data set and models studied. Additionally, the scope of the paper -- binary classification -- is not very broad.

---

> ### Author Response · Authors · 2022-11-14
> **Thank you for your review!**
>
> We really appreciate your review on our work!
>
> **"The paper considers a very limited set of neural network models that are not state-of-the-art. The paper can benefit from using a recent architecture to showcase the utility of the approach. The same applies to datasets. It is unclear why binary classification on MNIST is an interesting case to apply this technique to."** \
> While we only explore the binary classifiers, the backbone models, CNN and ResNet18, are modern architectures. The problem for binary classifiers with MNIST input is already very difficult. We have $256^{(28\times28)}$ samples to evaluate. This number is already much larger than the number of atoms in the universe. Our contribution is showing we don’t have to enumerate all of them but just a small number of them in order to get the correct output distribution! For ImageNet, this number is $256^{(224 \times 224 \times 3)}$ which is much much larger than we can afford without a supercomputer and thus we call for a more efficient sampler to achieve this task.
> People can resolve the issue of $10^{(400 \times 400)}$ size of configuration [1], but that’s because those physics models are relatively simple and people can take the advantage of just calculating the energy changed due to the one spin change. Our current DL framework does not support this. We should not be able to resolve this size of the configuration at this moment until we also have the DL framework support.
>
> Moreover, given our problem setting of considering the inputs in the entire input space, the binary classifier we consider can be applied to out-of-distribution detection which is also a binary classification. By using our techniques on the entire input space, we have already shown in sec 5.3 that the model under consideration is unlikely to distinguish the OOD samples and in-distribution samples for overconfident prediction logit regions. The output dimension is also extendable (see below) but the problem to be solved is more complicated.
> [1] "Scalable replica-exchange framework for Wang-Landau sampling," Vogel et al.
>
> **"It is not sufficiently clear how the new insights uncovered by applying this technique augment our understanding of neural networks. The paper can benefit from spelling that out more explicitly."** \
> Besides being mentioned as a future work in the introduction section, evaluation of the representative ratio of how many representative samples are recognizable to humans and how many of them are not in the relative positive logit regions has been demonstrated. In Sec. 5.3, Manual inspection on more representative samples, we sampled 200 representative samples in the relative positive logit region (should be confident to predict class 1) and we did not find any human recognizable samples (not even the human recognizable but wrongly predicted ones). Thus, we conclude in this human-evaluation of the unbiased representative samples as the model wrongly maps quite a large amount of samples to the relative positive logit region. This is not aligned with our human intuition and thus this model is not a good one. However, we do believe there must exist models that have a sufficient amount of human recognizable images mapped to this desired logit region.
>
> Here are the other practical utilities we haven’t done but also important (of course they are application dependent):\
> *Knowledge distillation.* When the two models are claimed to be the same, are their output distribution and the corresponding representative samples the same? \
> *Adversarial defense.* When adversarial defense algorithms are developed, do the representative samples and the output distribution improve? Does it lead to a more vulnerable model wrt other types of attacks?
>
> **"It is also unclear how this approach can be extended to the case of multiple labels. The paper can benefit from a discussion on that aspect as >2 label classification is more prevalent than a binary classifier."** \
> There already exist multi-dimensional output works [2]. One can follow our work here to revise using gradient-based proposals to efficiently sample multidimensional output density. Overall, the sampling problem setting is similar to one-dimensional, but since the joint output density is multi-dimensional, it takes much more time to converge. Thus, this paper introduced the idea of a "global" update (increase entropy S of some bins once by a certain amount) of the density once the bins have reached a certain threshold $S_{threshold}$ instead of accumulating the samples from scratch. This leads to faster convergence, but since it introduces more hyper-parameters to tune, we did not extend our method in this setting.\
>
> [2] “Wang-Landau algorithm for continuous models and joint density of states.” Zhou et al.

---

> ### Author Response · Authors · 2022-11-18
> **Could you please check and see if our revision will make our work meet the ICLR bar?**
>
> Thank you for your valuable comments and feedback! In the related work section, we compare the methods with MCMC samplers and elaborate (in the draft and in the follow-up of this review below) the sampling is indeed hard because the modern neural networks (we use both CNN and ResNet) are complicated compared to physics models. Binary classifiers can be used for a big family of problem OOD detection in our setting. In terms of the dataset we use, the MNIST is already intractable but we demonstrate this sampling is feasible. We believe an even more efficient sampler is needed for more complicated datasets.
>
> We position our method in terms of performance characterization literature (for comprehensive model evaluation). Our method is compatible with the previous performance characterization methods but offers the general sampling of the entire input space. Moreover, we update section 5.3 (and in appendix D) to present some of our interpretation of the results, such as why the black background appears when the logit gets larger. We also elaborate on how our methods connect to the previous works but provide new knowledge.
>
> Could you please check and see if our revision will make our work meet the ICLR bar? Really appreciate that!

---

> > ### Comment · Reviewer_ucC6 · 2022-11-21
> > **Thank you!**
> >
> > Based on the author's effort to address weak points raised in mine and other reviews, I will raise the rating by 1 point. Thank you!

---

### Official Review · Reviewer_KRCc · 2022-10-24

**Confidence:** 3
**Correctness:** 3
**Technical Novelty And Significance:** 3
**Empirical Novelty And Significance:** 4
**Recommendation:** 6

**Clarity, Quality, Novelty And Reproducibility:**

**Clarity:** To me, it is quite clear; I do think there are a few more things that I wish were on the paper. See W1 above, but the clarity is excellent for what is on the paper.

**Quality:** See strengths and weaknesses above.

**Novelty:** It is quite new and exciting!

**Reproducibility:** The analysis is built on existing tools, and clearly explained. While no code is promised, I think reproducibility is excellent.

**Strength And Weaknesses:**

**Strengths**

*[S1]:* Very interesting results: As mentioned in the summary above, the main result of this paper is quite interesting. The result can be potentially used to better understand adversarial attacks, out-of-distribution generalization, and learning mechanisms of neural networks in general. While the paper does not fully address the above possibilities, I remain hopeful :)

*[S2]:* Clearly written: The paper is very easy to read, despite this reviewer's unfamiliarity with either the "density of states" or the "Wang-Landau Algorithm"; After brief Wikipedia-based learning of these concepts, the paper made the parallels clear and I felt like I could map these concepts to things already known in ML literature and statistical properties. (Though I might have misunderstood something, hence noted confidence of 3 below).  Furthermore, the experiments are clearly written and well-explained.

**Weaknesses**

*[W1]:* Some obvious questions were left unanswered: For me, the paper was a little baffling due to how little analysis beyond the "main result" this paper goes into. Some questions that I would like to have seen discussed or at least mentioned (even if it is in "future works"):

W1.1. Why do most of the "human unrecognizable images" map to very negative logit values, instead of vice-versa; The positive-negative logit is normally considered arbitrary in binary classification (Two classes can be "assigned" to either side of them). Is there a reason for this asymmetry? What are its implications? Is one of the classes harder to create an adversarial attack and/or higher generalization than the other?

W.1.2 What are the implications of this finding in the context of decision-making, explainability, and "model evaluation? The abstract mentions that it will be explored in "future works" but in what manner? I am certainly not expecting this paper to do any of these "works" but I am quite unclear about how the findings from the paper can go beyond an interesting result and more towards how, in the words of the paper, "opens a new gate for neural model evaluation"

**Summary Of The Paper:**

The paper proposes to explore the daunting task of "modeling the entire space of input" to the output of a binary neural network. This is obviously impossible for any reasonably sized feature space. So, the paper proposes to do this by modeling the output of a neural network to the "density of states" concept in physics, where an efficient way to estimate it via the "Wang-Landau Algorithm", which seems to be a metropolis-hasting MCMC method. Wang-Landau algorithm can still yield many unrelated samples, so the paper proposes another recent work called Gibbs-With-Gradients (GWG) to be used as the sampler. GWG can leverage the model's learned weights to propose distributions that are closer to seen examples. Doing this exercise yields some interesting observations, the biggest of which is the fact that an enormous amount of possible inputs map to a very high-confidence prediction regime for a neural network.

**Summary Of The Review:**

I am currently moderately positive about the paper and the very interesting "main" finding of the paper. However, I refrain from a higher score because the discussion leaves a lot to be desired about whether this is a special case or can be used as a general-purpose analysis of any binary (or multi-class) classifiers.

---

> ### Author Response · Authors · 2022-11-14
> **Thank you for your recognition of our work.**
>
> Thank you for your recognition of our work!
>
> **"W1.1. Why do most of the "human unrecognizable images" map to very negative logit values, instead of vice-versa; The positive-negative logit is normally considered arbitrary in binary classification (Two classes can be "assigned" to either side of them). Is there a reason for this asymmetry? What are its implications? Is one of the classes harder to create an adversarial attack and/or higher generalization than the other?"**
>
> We believe that this result is consistent with the previous discovery, such as [1] that the model is using the background for classification. We further conducted another experiment that computes the L2 pixel-wise distance from the uniform noise image to the samples of class 1 and 0 respectively. The mean L2 distance from uniform noise to 0 is around 0.3121 and that from uniform noise to 1 is around 0.3236. The distance between 1 and 0 samples is 0.1652. This might provide some hints the model thinks uniform noise samples are closer to 0 samples than 1 samples and thus it is not symmetric.
> Again in this paper, we are not trying to interpret the results but to illustrate the fact that sampling the whole input space is feasible and it can produce interesting results to raise new research questions to understand the trained deep nueral network models.
> [1] “Noise or Signal: The Role of Image Backgrounds in Object Recognition,” Xiao et al.
>
> **"W.1.2 What are the implications of this finding in the context of decision-making, explainability, and "model evaluation? The abstract mentions that it will be explored in "future works" but in what manner? I am certainly not expecting this paper to do any of these "works" but I am quite unclear about how the findings from the paper can go beyond an interesting result and more towards how, in the words of the paper, 'opens a new gate for neural model evaluation'"**
>
> Besides being mentioned as a future work in the introduction section, evaluation of the representative ratio of how many representative samples are recognizable to humans and how many of them are not in the relative positive logit regions has been demonstrated. In Sec. 5.3, Manual inspection on more representative samples, we sampled 200 representative samples in the relative positive logit region (should be confident to predict class 1) and we did not find any human recognizable samples (not even the human recognizable but wrongly predicted ones). Thus, we conclude in this human evaluation of the unbiased representative samples as the model wrongly maps quite a large amount of samples to the relative positive logit region. This is not aligned with our human intuition and thus this could be an evidence this model is not a good one for OOD detection because of overconfident prediction. However, we do believe there must exist models that have a sufficient amount of human recognizable images mapped to this desired logit region.  :-)
>
> Here are the other practical utilities we haven’t done but are also important (of course they are application-dependent):\
> *Knowledge distillation.* When the two models are claimed to be the same, are their output distribution and the corresponding representative samples the same?\
> *Adversarial defense.* When adversarial defense algorithms are developed, do the representative samples and the output distribution improve? Does it lead to a more vulnerable model wrt other types of attacks?

---

> ### Author Response · Authors · 2022-11-18
> **The draft is revised to draw connection for model evaluation.**
>
> Thank you for your valuable comments and feedback! We update section 5.3 to present some of our interpretation of the results, such as why the black background appears when the logit gets larger. We also add the discussion of how our methods connect to the previous works but provide more knowledge. In the related work section, we compare the methods with MCMC samplers and position our method in terms of performance characterization literature (model evaluation).

---

### Official Review · Reviewer_JRJF · 2022-10-30

**Confidence:** 3
**Correctness:** 3
**Technical Novelty And Significance:** 2
**Empirical Novelty And Significance:** 2
**Recommendation:** 5

**Clarity, Quality, Novelty And Reproducibility:**

Clarity:  The paper is reasonably clear.
Quality:  The paper quality can be strengthened as the results are still preliminary.
Originality:  The paper combines well known elements in the sampling and physics literature. The problem addressed is a difficult one and the importance of the problem has been articulated in the past. Thus, the authors need to be commended for their work.
Reproducibility:  While adequate information is provided to implement the algorithm, it would be recommended to have an open source implementation with the ability to reproduce the experiments.

**Strength And Weaknesses:**

Strengths:
Offers the important point that one should explore the entire input space and the input's correspondence to output logits. Sampling methods used in physics literature are offered as a possible way to do the exploration efficiently instead of a brute force approach.

Weaknesses:
The emphasis that one has to explore the relationship between the entire input space and output space is well known (see for example: performance characterization literature in the 90's,  http://haralick.org papers on performance characterization. There were explorations involving mapping of boolean random series to outputs for nonlinear operators (e.g. mathematical morphology).  Another example of such characterization in pattern classification settings is illustrated in Gao et al, Statistical Characterization of Morphological Operator Sequences in ECCV 2002).  However, the results in the present paper are not yet convincing to me - I find the visualizations in figures 2, 3, 5, 6 hard to judge. While there is illustration in the toy example that the algorithm seems to be performing correctly via quantitative overlay of the enumeration (groundtruth) against the paper's method, the practical utility of the methodology to gain insights is still not clear. Moreover, there are open challenges with respect to getting the sampling scheme to be efficient as rightly articulated by the authors.
Apart from the fact that the exploration of the input space is shown in the paper, the interpretation of the results is vague.



**Summary Of The Paper:**

The essence of the paper is that one should examine the behavior of a neural network by studying how the output logits change as a function of the entire input space rather than only examine the behavior based on the sampled data distribution. The paper then illustrates how one could utilize intuition regarding a correspondence between state density representations in physics to histograms of logits to design a sampling scheme that combines wang-landau algorithm with gradient with gibbs sampler to explore the vast space of output possibilities.  The algorithm is applied to a toy data set with reduced dimensionality (of 25 dimensions) constructed out of MNIST to illustrate the utility and inner working of the concept. This is then followed by tests on MNIST and illustration of certain insights (e.g. how negative logits correspond to human unrecognizable images and about the nature of what information may be used by the neural net).  Open issues in the sampler are discussed and it is articulated that this form of extensive exploration of the energy density is important for understanding neural net behavior in the entire input space of possibilities.

**Summary Of The Review:**

I like the overall idea of the paper and believe that complete characterization of the input to output behavior, over the entire input space, for a trained net will be invaluable for providing guarantees of performance.  However, in my view, the paper in its present form is still not ready for publication.

After the revisions presented by the authors I have increased my original rating and have presented my viewpoints to the reviewers and area-chair.  The paper has been strengthened and is addressing an important aspect of performance analysis.  After my discussion with the other reviewer and area-chair my revised rating does not change.

---

> ### Author Response · Authors · 2022-11-14
> **Thank you for your review!**
>
> **"The emphasis that one has to explore the relationship between the entire input space and output space is well known (see for example: performance characterization literature in the 90's,  http://haralick.org papers on performance characterization. There were explorations involving mapping of boolean random series to outputs for nonlinear operators (e.g. mathematical morphology). Another example of such characterization in pattern classification settings is illustrated in Gao et al, Statistical Characterization of Morphological Operator Sequences in ECCV 2002)."** \
> It is encouraging for us to see similar problems have been explored in the early days. We sincerely appreciate the reviewer for pointing this out. We also cited other works [1,2] that tried to understand models that approximate a pre-known function. We will cite the related work and use them to emphasize the importance of considering the entire input space.
>
> We read the referred paper [3] on characterizing the statistics of mathematical morphology. Whereas the previous work is trying to understand the input-output relations for better selecting the operators and their parameters, our work is trying to characterize the relationship of many inputs to output, given the model is trained (parameters are fixed). The motivation is substantially different because our inputs do not necessarily in the training set. The idea behind this is even though a model, which can also be a mathematical morphology operator, can reduce the error rate, the model can still make mistakes when the inputs do not distribute as the training samples. Besides, though the mathematical morphology is characterizing the multi-dimensional output, our model under consideration is much more general and complicated than a non-linear morphological operator.
>
> We’d like to emphasize that understanding the input-output relationship of the modern, deep neural network models is more challenging because of the architectures and the applications it can be applied to. For these deep neural network models, it is almost impossible to obtain the explicit formula for the output distribution’s density function for the inputs in the entire input space. We instead draw the connections from neural networks to physics and further revise the Wang-Landau algorithm to make it efficient and effective for our problem. As recognized by reviewer gv1q, our work is groundbreaking, offering novel tools to investigate and evaluate neural networks in a more comprehensive manner. \
>
> **"Moreover, there are open challenges with respect to getting the sampling scheme to be efficient as rightly articulated by the authors. "** \
> We agree that the sampling scheme can be improved so that it can be used for larger scale problems.
>
> **"Apart from the fact that the exploration of the input space is shown in the paper, the interpretation of the results is vague."** \
> We try to present a first tool to sample for output space for modern neural networks as a whole which are much more complicated than the operators as in the mathematical morphology [1]. We demonstrate that this is feasible and efficient compared to previous works as we demonstrated in fig (2-5). It can also generate some interesting results (fig 3-6). For example, we have strong evidence that the CNN model maps many OOD samples to overconfident prediction logit regions.
> There is a trend where darker pixels of representative samples are preferred when logit gets larger. Even though we observe these results, it is not the goal of the paper to interpret them because understanding what a model is thinking requires systematic and sophisticated experimental designs and hypothesis testing. It is beyond the scope of this paper. Follow-up research on interpretation of our results (and new results) are what we expect in the future work.
>
> [1] “Determining functional relationships from trained neural networks.” Hammitt et al. \
> [2] “Mapping the energy landscape.” Barbu et al. \
> [3] “Statistical Characterization of Morphological Operator Sequences.” Gao et al.

---

> > ### Author Response · Authors · 2022-11-14
> > **Additional comment for addressing the weakness**
> >
> > **"However, the results in the present paper are not yet convincing to me - I find the visualizations in figures 2, 3, 5, 6 hard to judge. While there is illustration in the toy example that the algorithm seems to be performing correctly via quantitative overlay of the enumeration (groundtruth) against the paper's method, the practical utility of the methodology to gain insights is still not clear. "** \
> > Given our problem setting of considering the inputs in the entire input space, our method can be applied to out-of-distribution (OOD) detection. It has been well-known that the model predicts a lot of unrecognizable samples to high confidence regions (very high prediction probability). Our output distribution results echo this observation and further supplement more findings: (1) high prediction probability issue might be more severe in one class; (2) paired with human evaluation, we can understand the ratio of the highly confident unrecognizable samples and the in-distribution samples as shown in Sec. 5.3, Manual inspection on more representative samples. One implication for this result is if we use the models that we tested with our sampler in our experimental section for OOD detection with likelihood-based scores, such as energy score [1] and max-softmax score[2], may not lead to good performance for generic types of OOD samples.  \
> >
> > Here are the other practical utilities we haven’t done but also important (of course they are application dependent):\
> > *Knowledge distillation.* When the two models are claimed to be the same, are their output distribution and the corresponding representative samples the same? \
> > *Adversarial defense.* When adversarial defense algorithms are developed, do the representative samples and the output distribution improve? Does it lead to a more vulnerable model wrt other types of attacks?
> >
> > [1] “Energy-based Out-of-distribution Detection.” Liu et al.\
> > [2] “A baseline for detecting misclassified and out-of-distribution examples in neural networks.” Hendrycks et al.

---

> > > ### Comment · Reviewer_JRJF · 2022-11-14
> > > **I did not get a direct answer to my question..**
> > >
> > > My concern is that the visualizations presented in your figures were hard to judge.  My concern was not about the usefulness for detection of OOD or about the fact that CNN models provide overconfident logits. Unfortunately, I think your argument diminishes your main message, as  OOD detection can be done by many other methods in the literature.  My recommendation is to stick to your justification on the fact that you have leveraged the correspondence between state density representations in physics to histograms of logits to explore the behavior of the deep net.  I acknowledge that this aspect of your paper is valuable. The main concern I have is that with a convincing demonstration of the strengths of your approach requires a revision of your paper.

---

> > > > ### Author Response · Authors · 2022-11-15
> > > > **May we have more ideas of what strengths are you referring to?**
> > > >
> > > > Thank you so much for your comments. May we have more ideas of what strengths are you referring to?
> > > >
> > > > We will add Gao’s work to our related work. As Gao’s method has the potential “if the appropriate embedded markov chain is constructed,” are there any follow-up works with this method with the “appropriate Markov chain” you mentioned that is applied to modern deep networks but we happened to miss them? In fig 4, we have shown applying the Wang-Landau algorithm to an input space with $256^{28 \times 28}$ (not the binary pixels as Gao et al. considered in their paper) is not possible. The sampler correctness is demonstrated in fig 2 and its efficiency is demonstrated in fig 4.
> > > >
> > > > Indeed, there are many other OOD detection methods, including the likelihood methods. Our finding provides some evidence that challenges the likelihood-based OOD detection methods that may be more model-dependent than we thought before: if the model cannot map most of the human unrecognizable samples to around 0, the likelihood-based OOD detection methods cannot do much for OOD detection for samples in the entire input space. (This does not exclude the possibility of good performance for task-specific OOD samples.) The implication of fig3 and 5 is that some models (the CNN and ResNet tested) are not going to work well for OOD detection with likelihood-based OOD detection methods if they are exposed to the samples within the uniform sampling space because most of the samples will be mapped to high confident prediction regions. For CNN, even the in-distribution regions (red lines) do not uniquely correspond to in-distribution samples. We show this by counting 200 samples in those in-distribution regions (fig 6) but we do not find one human-recognizable sample. More importantly, whereas previous works only showed the existence of the overconfident prediction samples [1], we know the relative counts between them vs the expected in-distribution logit regions (for CNN): the most dominant region (logit=-55) is around 3100 vs 1 (logit=13) in log scale and the overconfident regions (logit=-55) do not correspond to samples with class=0 at all. This can give us a rough idea of how this specific CNN performs and we can tune the model again and sample these output distributions to confirm if the model improves.
> > > >
> > > > If you have other strengths we may miss but you think it is important to demonstrate, please let us know. Thank you for your valuable comments!
> > > >
> > > > [1] “Deep neural networks are easily fooled: High confidence predictions for unrecognizable images.” Nguyen et al.

---

> > > > > ### Comment · Reviewer_JRJF · 2022-11-15
> > > > > **Clarification of my comments**
> > > > >
> > > > > My feedback about applicability of Gao et al's approach to modern settings is mainly an observation. As far as I know, there is no one who has extended that work to neural net settings.  There are recent efforts at perturbation propagation through neural networks. See for example: Daniel A. Roberts et al, The Principles of Deep Learning Theory: An Effective Theory Approach to Understanding Neural Networks.
> > > > >
> > > > > Thank you for elaborating on the details of the figures.  My concern about your paper is not about your methodology but rather whether the paper in its present written form is good enough for publication.  Revisions, addressing the broader link to past literature and to highlight the key strengths and limitations of your method, will strengthen the paper.

---

> > > > > > ### Comment · Reviewer_JRJF · 2022-11-15
> > > > > > **Another point about relationship to past work**
> > > > > >
> > > > > > I hope I can add a few points that may further provide some insights:
> > > > > >
> > > > > > The essence of the performance characterization work in computer vision, advocated by Haralick, Binford, Foerstner and others done in the 90's and 2000's, is the observation that meta-analysis of a human-expert specified system should be analyzed in specific application contexts.  For a reference on systems analysis in the context of a real-world application please see:  Greiffenhagen et al (2000 to 2001),  e.g. https://ieeexplore.ieee.org/document/959343 .  A review paper on the topic is presented in Thacker et al (2008) - https://www.sciencedirect.com/science/article/abs/pii/S1077314207000793 .  There is connection to uncertainty quantification (see classical references in numerical methods literature in applied mathematics - https://en.wikipedia.org/wiki/Uncertainty_quantification), Bayesian methods/ Probabilistic programming (https://en.wikipedia.org/wiki/Probabilistic_programming: see numerous references from late 2000's from Tenenbaum and Collaborators at MIT), and Probabilistic numerics (see recent book by Hennig et al) that are now pursued in the literature.  In summary, performance analysis is relational and various efforts have been done in past based on modeling and systems perspectives.  Theoretical characterization of systems is known to be hard and computational tools are necessary.
> > > > > > Efficient sampling for exploration of systems performance is an open research topic. Your work adds to this body of knowledge.

---

> > ### Comment · Reviewer_JRJF · 2022-11-14
> > **Context of previous research and contrast with present work is not yet clearly stated**
> >
> > The central aspect of performance characterization, as discussed in the 90's by haralick and his collaborators, is to characterize the input to output distributional relationships across nonlinear sequence of parametrized operators.  The challenge, as articulated precisely by Haralick, is in doing this for a wide range of input probability distributions.   The work of Gao et al has two steps - a step that involves explicit calculation of probability mass function of discrete random variates of interest in the output as a function of a probability distribution in the input, nonlinear operator choices, and their parameters.  The computational method is not specific to a given input data distribution as the input data distribution can be varied to be any form.    Note that Gao et al does not provide closed form solutions but rather provide a numerical method of explicit computation of the output statistics for a given choice of input statistics and operator sequence.  Thus it may be applied to modern neural network settings if the appropriate embedded markov chain is constructed. However, I do agree that there are differences between your method and what Gao et al's or Haralick's past works.  The formal distribution propagation is indeed rather involved.  Indeed, there has been analysis for deep learning - e.g. https://arxiv.org/abs/2106.10165 .  It is precisely because of the complexity of performance characterization that modern practices advocate sampling as a mechanism.
> >
> > The fact that CNN models map OOD data with overconfident logits is rather well known, and I acknowledge that your methodology verifies the sam.

---

> ### Author Response · Authors · 2022-11-18
> **Could you please check and see if our revision will make our work meet the ICLR bar?**
>
> Thank you for your valuable comments and constructive discussion! In the related work section, we add the previous works we believe that are most relevant (performance characterization and uncertainty propagation). We also characterize our own experimental setting using the paradigms proposed in the performance characterization. Thanks for your suggestion, the close relationship to performance characterization is not missed.
>
> In the experiment section, we update section 5.3 (and in appendix D) to present some of our interpretation of the results, such as why the black background appears when the logit gets larger. We also elaborate on how our method connects to the previous works but provides new knowledge. Could you please check and see if our revision will make our work meet the ICLR bar? Really appreciate that!

---

> > ### Comment · Reviewer_JRJF · 2022-11-18
> > **Change of rating**
> >
> > Thank you for your efforts to link your work and set it in the context of the background literature.  In light of this, I change my rating by two points.  Furthermore, I will take a close look at your answers to other reviewers questions and see if it warrants a further change in rating.

---

> > > ### Author Response · Authors · 2022-12-11
> > > **Thank you for your feedback**
> > >
> > > Thank you for your recognition of our edition of the paper. Moreover, we found the other reviewers have already accepted the revision that resolved their concerns. Would you mind checking the comments? Thank you so much for your time!

---

### Official Review · Reviewer_gv1q · 2022-11-04

**Confidence:** 3
**Correctness:** 4
**Technical Novelty And Significance:** 3
**Empirical Novelty And Significance:** 4
**Recommendation:** 6

**Clarity, Quality, Novelty And Reproducibility:**

The articles are clearly written and of good quality.

The use of a physical perspective for interpretable analysis of the mapping of neural networks is novel.

**Strength And Weaknesses:**

**[Strength]**

- The idea of trying to understand neural networks through input-output mapping relationships is groundbreaking; it also makes sense to draw on solutions from a related field (condensed matter physics).
- Various experiments well illustrate that the statistics sampled by the proposed method (GWL) are close to the ground truth (enumeration) and significantly better than directly sampling (from in-dist test samples).

**[Weaknesses]**

- While it is possible to describe the output histogram problem using a domain specific approach (physical language), I still wonder if there is another way (perhaps a more general sampling algorithm) to solve this problem? The authors do not seem to discuss this sufficiently in the Related Work section.
- Are the properties of neural network input-output mapping observed by the authors in their experiments (e.g., mapping unrecognizable human samples to very negative logit values) consistent or contradictory to some previous literature? Making some comparisons would make the overall analysis a bit more comprehensive.
- Fig. 6 is interesting, but I don't particularly understand why logit values for larger samples look "darker" implying that the *background* is more important than the *foreground* for the classification task? Perhaps the authors could give a more detailed explanation on this?



**Summary Of The Paper:**

This paper attempts to analyze a binary neural classifier's input-output relationship.
A straightforward, brute-force sampling is forbidden due to the high dimension and continuity of input distribution.
To achieve efficiency, the authors reformulate the problem as density of states (DOS) sampling in physics, and propose to apply a gradient-based proposal which facilitates the exploration of under-explored output values.
The main contributions are:

- making the output histogram problem tractable.
- analyzing the input-output relationship in the entire input space and revealing some interesting properties by sampled statistics.
- introducing a new perspective to understand the behavior of neural network classifiers.


**Summary Of The Review:**

In general I think it is novel and interesting to use a physical perspective to understand the mapping of neural networks. However, there is very little discussion or comparison of related work throughout the paper (e.g., other non-physically inspired sampling algorithms, or other work on the analysis of neural network mappings; see the Weakness section above). This results in this work looking less than complete, and weaken their conclusions. I am inclined to marginally reject it.


========== post rebuttal ==========

> We update section 5.3 (and in appendix D) to present some of our interpretation of the results, such as why the black background appears when the logit gets larger.
> In the related work section, we compare the methods with MCMC samplers and position our method in terms of performance characterization literature.

Thank you for clarifying this and adding Section 5.3 as well as the supplementary material for more explanation. I feel that the two most significant flaws (inadequate discussion of the related work and too few explanations of the Fig. 6) have been addressed, thus improving my rating.

---

> ### Author Response · Authors · 2022-11-14
> **Thank you for your recognition of our work.**
>
> We really appreciate your recognition of the groundbreaking nature of our work. We sincerely believe that our sampler opens the gate toward a more comprehensive evaluation of deep neural networks.
>
> **"While it is possible to describe the output histogram problem using a domain-specific approach (physical language), I still wonder if there is another way (perhaps a more general sampling algorithm) to solve this problem. The authors do not seem to discuss this sufficiently in the Related Work section."** \
> We will add more discussions about alternative samplers in related work. There are other samplers that can resolve this issue, but most of them are also Wang-Landau variants because of the following reasons. First of all, the previous sampling is based on the function or model that we can sample from, such as the formula for the Ising model or a neural network’s energy function for energy-based-models, whereas the fact that we do not have formulas or models for the output distribution which we are trying to sample from really limits our choice of the MCMC samplers. If we want to do better than uniform sampling, we have to use the recorded histogram (or other measures that can record the current approximation of the distribution) to drive the sampler to explore the whole distribution. Secondly, for a trained model, what we are trying to sample is believed to have non-trivial output regions, such as those that are likely (only) to correspond to the in-distribution samples. Given the complexity of the model, there may be output barriers that the other samplers generally take much longer or even produce completely wrong results (given some undesired initial configurations). The Wang-Landau algorithm, on the other hand, is able to overcome the barriers to sampling the whole distribution. \
> The major goal of this paper is to show that sampling the entire input space is possible. After we draw the connection to physics, the Wang-Landau algorithm becomes the most intuitive and generic solution. There are many more modern Wang-Landau variants, but we only adopt the original one to prove that sampling the entire input space is feasible and efficient. As future works, we can use modern Wang-Landau algorithms for multi-dimensional outputs [1] and parallelization [2].
> [1] “Wang-Landau algorithm for continuous models and joint density of states.” Zhou et al.  \
> [2] “A practical guide to replica-exchange wang—landau simulations.” Vogel et al.
>
> **"Are the properties of neural network input-output mapping observed by the authors in their experiments (e.g., mapping unrecognizable human samples to very negative logit values) consistent or contradictory to some previous literature? Making some comparisons would make the overall analysis a bit more comprehensive."**
> It has been well-known that the model predicts a lot of unrecognizable samples to high confidence regions (very high prediction probability). Our output distribution results echo this observation and further supplement more findings: (1) high prediction probability issue might be more severe on one class than others; (2) paired with human evaluation, we can understand the ratio of the highly confident unrecognizable samples and the in-distribution samples. One implication for this result is if we use the models that we tested with our sampler in our experimental section for OOD detection with likelihood-based scores, such as energy score [1] and max-softmax score [2], may not lead to good performance for generic types of OOD samples.\
> [1] “Energy-based Out-of-distribution Detection.” Liu et al.\
> [2] “A baseline for detecting misclassified and out-of-distribution examples in neural networks.” Hendrycks et al.

---

> > ### Author Response · Authors · 2022-11-14
> > **Addressing the last weakness comment**
> >
> > **"Fig. 6 is interesting, but I don't particularly understand why logit values for larger samples look "darker" implying that the background is more important than the foreground for the classification task? Perhaps the authors could give a more detailed explanation on this?"**
> > We conjecture that this result is consistent with the previous discoveries, such as [1] that the model is using the background for classification. We suspect that the model considers the unrecognizable samples closer to 0 than 1. We conducted another experiment that computes the L2 pixel-wise distance from the uniform noise image to the samples of class 1 and 0 respectively. The mean L2 distance from uniform noise to 0 is around 0.3121 and that from uniform noise to 1 is around 0.3236. The distance between 1 and 0 samples is 0.1652. This might provide some hints the model thinks uniform noise samples are closer to 0 samples than 1 samples. \
> > Our main contribution is to illustrate the fact that sampling the whole input space is feasible and our preliminary analyses on the results obtained by the sampler already show interesting findings. We expect there will be follow-up works that can deliver a more comprehensive analysis of the results. \
> > [1] “Noise or Signal: The Role of Image Backgrounds in Object Recognition,” Xiao et al.

---

> ### Author Response · Authors · 2022-11-18
> **Could you please check and see if our revision will make our work meet the ICLR bar?**
>
> Thank you for your valuable comments and feedback! We update section 5.3 (and in appendix D) to present some of our interpretation of the results, such as why the black background appears when the logit gets larger. We also elaborate on how our methods connect to the previous works but provide new knowledge. In the related work section, we compare the methods with MCMC samplers and position our method in terms of performance characterization literature. Could you please check and see if our revision will make our work meet the ICLR bar? Really appreciate that!

---

### Decision · Program_Chairs · 2023-01-20

**Decision:**

Reject

**Justification For Why Not Higher Score:**

 It would be reasonable to accept this paper, if value was to be placed mainly on the conceptual contributions.  However, it was also felt that additional experimentation, comparison, and situation within existing methods in ML would be valuable here.

**Justification For Why Not Lower Score:**

NA

**Metareview: Summary, Strengths And Weaknesses:**

Thank you for your submission to ICLR.  Overall, this submission received a very large amount of discussion, both within OpenReview and within the AC-reviewer meeting held on this paper.  As such, I will summarize the basic understanding of the paper as it currently stands, highlighting the positives and negatives.

On the positive side, this paper considers an interesting and potentially valuable contribution to the field: trying to better understand how networks react to the "entire" distribution over input space, to see what kind of distributional properties exist in the outputs. While (as reviewers pointed out, and as the authors acknowledged in their responses), this is by no means a truly new perspective, the paper nonetheless tackles some important problems in this space and brings to bear interesting techniques (namely, the Wang-Landau algorithm) to this problem.  Naturally, understanding the _full_ distribution of possible outputs over a high-dimensional input space represents an intractable problem, the approaches here cast a new perspective on this challenge, one which may be valuable to researchers going forward.

The main downside to this work, however, is simply that it was unclear to the reviewers and myself "just how well" the proposed approach really works at casting insight into this problem.  Specifically, the authors ultimately propose a method based upon the Wang-Landau, essentially a particular form of MCMC estimator with gradient information, to compute the necessarily integrals over input space.  However, there is very little comparison of the approach to alternative, and similar, MCMC methods that are more widely used within the ML space, such as Hamiltonian MC methods (see e.g. https://arxiv.org/pdf/1701.02434.pdf).  The experimental results don't really compare the proposed method to alternative MCMC approaches, or fully illustrate the consequences of this approach is a manner that substantially challenges the common knowledge of adversarial inputs.

For these reasons, the ultimate feeling is that the paper is not quite ready yet for publication, and would benefit substantially from additional experimentation, evaluation, and situation in contrast to alternative approaches within the field.



**Summary Of Ac-Reviewer Meeting:**

 The AC/reviewer meeting largely centered around the above discussion of the paper, highlighting the positive elements of the paper (considering a challenging and important problem within the field) plus the negative elements (the fact that the actual solution methods were of unclear empirical quality).  The discussion actually tended towards a more positive view of the paper than was initially clear from the reviews, and it was repeatedly emphasized that there was indeed a valuable conceptual contribution here.  But there was also the acknowledgement that the paper likely didn't go far enough from the standpoint of actual evaluation, for a paper that ultimately did hinge on providing new experimental perspectives on deep learning.